# Characteristics of Future Models of Integrated Outpatient Care

**DOI:** 10.3390/healthcare7020065

**Published:** 2019-04-27

**Authors:** Alan Leviton, Julia Oppenheimer, Madeline Chiujdea, Annalee Antonetty, Oluwafemi William Ojo, Stephanie Garcia, Sarah Weas, Eric Fleegler, Eugenia Chan, Tobias Loddenkemper

**Affiliations:** 1Division of Epilepsy and Clinical Neurophysiology, Department of Neurology, Boston Children’s Hospital and Harvard Medical School, 300 Longwood Avenue, Boston, MA 02115, USA; julia.e.oppenheimer@gmail.com (J.O.); Madeline.Chiujdea@childrens.harvard.edu (M.C.); Annalee.Antonetty@childrens.harvard.edu (A.A.); William.femi@gmail.com (O.W.O.); stephanie.liz.garcia@gmail.com (S.G.); Tobias.Loddenkemper@childrens.harvard.edu (T.L.); 2Division of Developmental Medicine, Department of Medicine, Boston Children’s Hospital and Harvard Medical School, 300 Longwood Avenue, Boston, MA 02115, USA; Sarah.Weas@childrens.harvard.edu (S.W.); Eugenia.Chan@childrens.harvard.edu (E.C.); 3Division of Emergency Medicine, Department of Medicine, Boston Children’s Hospital and Harvard Medical School, 300 Longwood Avenue, Boston, MA 02115, USA; Eric.Fleegler@childrens.harvard.edu

**Keywords:** consumer health informatics, E-health, mobile apps, outpatient follow-up, patient portal, patient-reported outcomes

## Abstract

Replacement of fee-for-service with capitation arrangements, forces physicians and institutions to minimize health care costs, while maintaining high-quality care. In this report we described how patients and their families (or caregivers) can work with members of the medical care team to achieve these twin goals of maintaining—and perhaps improving—high-quality care and minimizing costs. We described how increased self-management enables patients and their families/caregivers to provide electronic patient-reported outcomes (i.e., symptoms, events) (ePROs), as frequently as the patient or the medical care team consider appropriate. These capabilities also allow ongoing assessments of physiological measurements/phenomena (mHealth). Remote surveillance of these communications allows longer intervals between (fewer) patient visits to the medical-care team, when this is appropriate, or earlier interventions, when it is appropriate. Systems are now available that alert medical care providers to situations when interventions might be needed.

## 1. Introduction

The traditional system of medical care is characterized by visits to the clinician who asks about the recent past, examines the patient, makes decisions about therapy, and schedules a follow-up assessment. Between visits, patients often have little contact with their clinical-care team.

The multiple initiatives to encourage innovations in medical-care delivery, which both enhances the quality of care and reduce costs, have allowed a considerable flexibility [1,2,3,4,5]. Here, we offer a ground-level perspective of clinicians and other care providers who are preparing for the soon-to-be reality of outpatient care of children and adults with chronic diseases.

This emerging system of care delivery incorporates greater emphasis on self-management (e.g., consuming medications as prescribed, deciding about the need for medical-care or other adjustments), electronic patient-reported outcomes (i.e., symptoms) (ePROs), the “remote surveillance” system that receives these ePROs and electronic data collected by sensors (e.g., glucometers, activity monitors, internet enabled scales), assistance navigating service systems, and alerting programs and algorithms that inform the clinical-care team when additional assessment and intervention might be needed [6]. In this overview, we discuss each of these elements, first individually, and then discuss the systems that integrate them.

## 2. Why the Need for the New System?

The traditional system of periodic outpatient assessments relied on fee-for-service, wasted resources when patients were well (with unnecessary outpatient visits), underutilized resources when care was needed between clinic visits [7], and prompted some patients or their surrogates to delay seeking care for an adverse event, especially outside the bounds of the regular doctors’/clinic hours [8]. In the new system, in keeping with the Affordable Care Act, the fee-for-service model is replaced with capitation arrangements accepted by accountable care organizations, putting pressure on clinical-care teams and their organizations to justify that what they do is the most cost-effective option [9]. Altering care delivery patterns can reduce costs [3], but not invariably [10].

## 3. Characteristics of the New System

### 3.1. Self-Management 

Self-management has been defined as “an individual’s ability to detect and manage symptoms, treatment, physical and psychosocial consequences, and lifestyle changes inherent in living with a chronic condition” [11]. Others suggest that self-management of medical conditions involves three tasks—medical management, role management, and emotional management [12]. 

Also defined as an integrated coordination of healthcare interventions and actions for populations with chronic conditions, self-management “supports the physician or practitioner/patient relationship and plan of care; emphasizes prevention of exacerbations and complications through the use of evidence-based practice guidelines and patient empowerment strategies; and evaluates outcomes on an ongoing basis with the goal of improving overall health” [13]. We parse this sentence beginning with “integrated coordination supports the physician or practitioner/patient relation and plan of care.”.

The Federal Health IT Strategic Plan: 2015–2020 has as its first goal, “Advance Person-Centered and Self-Managed Health” [14]. This is to be achieved by empowering the individual, the family, and the caregiver health management and engagement.

The coordination of care among multiple clinicians, results in what has come to be called “family-centered care” [15,16,17]. Emphasizing “patient empowerment strategies,” this approach encourages patients and caregivers to participate in the care [18,19,20], fosters medication adherence [21], helps family members cope [22], and increases self-esteem, and quality of life [23]. 

Engaging the patient and her family can also be enhanced with assistance navigating service systems [24]. Programs that do this, appear to improve coping and increase the use of community-based services and resources. Parent education programs also improve parents’ mental health, as well as communication and problem-solving skills [25].

We continue parsing the introductory sentence and focus now on “evaluates outcomes on an ongoing basis.” Although repeatedly completing forms that require patients or care-givers to identify problems, rate their magnitude, and identify their priority might seem like a burden, some families have felt that using the ePROs had several potential benefits, including an aid to remember something they wanted to discuss with the clinical-care team, and making the next consultation more efficient and patient-centered [26]. (Please see the next section on remote surveillance for more on this topic.)

Self-management includes all the activities that patients and their families do, to increase health-related quality of life, such as identifying and avoiding triggers that exacerbate symptoms, and adherence to prescribed medications and other therapies [27]. Questionnaires not only assess, but can also expand the self-management knowledge [28,29,30,31,32].

Often patients and their families need help with self-management [11,33]. “Supporting self-management is about helping patients to develop skills such as problem solving, setting goals, accepting change, finding coping strategies, managing relationships through communication, and finding quality of life in difficult circumstances” [34]. By providing support, physician and nurse clinicians [35,36], and pharmacists [37] can help people acquire each of these skill sets, as well as facilitate self-management in other ways [38]. Unfortunately, clinicians do not always have sufficient training, time, resources, or appropriate skills or confidence to provide effective self-management support [39,40]. 

Multiple techniques are used to maximize patient/family engagement in self-management, including increased support from clinicians [35], education about the person’s disease and available self-management resources [28,29,30,31,32,39], prompts to increase shared decision-making with the clinicians [41], and customizing the available options [42,43]. 

What is needed to engage patients and their families appears to be influenced, in part, by the needs posed by the medical condition. For example, the difficulties some parents experience in “navigating service systems, finding information about their child’s condition, and accessing health care and community resources” can be minimized by having “interventions that “activate” parents of children with special health care needs to increase their knowledge, skills, and confidence in managing, coordinating, and advocating for their child’s needs” [24].

While most measurements of the success of self-management focus on medication adherence, the need exists for improved assessments of symptom reduction capabilities [44]. Self-management programs have proven successful in improving the attainment of goals for the care of diabetes and hypertension [45]. They also hold promise for the self-management of asthma [46]. Visualization of symptom reduction by the patients and their families could further reinforce self-management by providing feedback and a greater sense of control.

### 3.2. Electronic Capabilities

One of the early clearer definitions of e-health is “the combined use of electronic communication and information technology in the health sector” [47]. Nevertheless, a decade later, one author claimed that no definition is yet well-accepted [48]. At about that time, The World Health Organization (WHO) defined e-Health as ‘the transfer of health resources and healthcare by electronic means’ [49]. 

In another report, the World Health Organization defined mHealth as “the use of mobile and wireless technologies to support the achievement of health objectives” [50]. Sensors (both wearable, and in place) hold the promise of providing clinicians with previously-unavailable information that might even be lifesaving [51]. A total of 80% of adults with an implanted cardioverter defibrillator or cardiac resynchronization device accept remote monitoring [52], which appears to reduce in-hospital visit numbers, time required for patient follow-up, physician and nurse time, and hospital and social costs [53].

“To deliver better health care at a lower cost, health information technology (IT) should be redesigned to support improved, patient-centered care, and not the isolated tasks of physicians and clinicians” [54]. Smart phone apps have the potential to improve self-management interventions [55,56]. The most common features of these apps are “real-time or frequent periodic symptom assessments, pre-programed reminders, and feedbacks tailored specifically to the data provided by participants” [57].

Some systems now collect physiological data remotely, using “off-the-shelf” easy-to-use components geared towards the sports or entertainment market [58]. Although some of the commercially available apps appear to have the potential to promote self-management [59,60], very few studies have integrated physiological data with health care personnel communication [61]. 

Mobile phone messaging applications, such as Short Message Service (SMS) and Multimedia Message Service (MMS), can provide medication reminders, therapy adjustments, and supportive messages. Although they, thus, offer the promise of supporting self-management, the evidence that they help achieve these goals remains limited [62,63]. The evidence is accruing, however, that mobile technologies can help treat heart failure [64,65,66], hypertension [66,67], asthma [68,69,70], depression [71], diabetes [72,73,74,75,76], overweight, and obesity [77], and also enhance medication adherence [78,79].

Both data encryption and data quality requirements differ for medical care and recreational device use [80]. In addition, barriers to the acceptance and spread of mHealth are abundant [81]. Nevertheless, mHealth and related technologies appear to be increasingly accepted [82,83,84].

The effectiveness of mHealth technologies to relay critical information in real-time, apparently depends on the strength of the communication loop between patients and health care professionals [85]. Patient (or caregiver)-provided or sensor-provided data can be sorted by level of urgency defined by customized algorithms and be distributed to the most appropriate recipient. Only non-physician members of the health team screen communications about changes from the status quo that are deemed modest and non-critical. Values beyond the critical thresholds defined for each patient, according to the physicians’ preference, trigger alerts that are sent both to the patient’s non-physician clinical team and her/his physician. Messages labeled as urgent or high priority by the patient or caregiver can also be disseminated among the non-physician clinical care team members and physicians.

The use of information and communication technologies to deliver healthcare at a distance, also supports “patient self-management through remote monitoring and personalised feedback” [86]. These appear to be especially effective for adults with severe chronic diseases, who are at a high-risk of hospitalization and death [86].

The term “patient portal” also has had multiple definitions [87,88]. The federal government uses the term “patient portal” for a secure web site, integrated with the electronic health record (EHR), that allows the patient to complete forms, access personal health information such as problem lists, current medications, immunization history, laboratory data, and radiology reports, schedule appointments, and request prescription refills [89]. These portals, which allow patients electronic access to appointment scheduling, medication refills, and secure communication with their provider or care team, seem to improve “medication adherence, disease awareness, self-management of disease, a decrease of office visits, and an increase in disease-prevention behaviors” [90]. 

Although initially slow, acceptance of patient portals is now increasing [91]. Teaching patients how to access the portal appears to increase the portal use, as does providing answers to questions about information obtained via the portal [91].

Some patient portals also allow electronic communication with healthcare providers, although this feature is often described as secure messaging, which the Centers for Medicare and Medicaid Services (CMS) define as “any electronic communication between a provider and patient that ensures only those parties can access the communication. This electronic message could be email or the electronic messaging function of … an online patient portal, or any other electronic means” [92]. The name “eVisits” has been applied to structured communications about symptoms [93].

The messages are encrypted and integrity-protected, in accordance with the standards for encryption and hashing algorithms. In the United States, any electronic communication between patient and provider needs to conform to the regulations in the Health Insurance Portability and Accountability Act (HIPAA) [94].

Acceptance of secure messaging by patients was initially slow, but has since accelerated [95]. Secure messaging has the potential to improve medication adherence and clinic attendance [96], increase patient satisfaction, be accepted as convenient, time-saving, and useful, as well as reduce the costs for the patient [97]. On the other hand, these benefits have come with increased burdens on healthcare providers and staff [98], and by extension—at least initially—apparently increased costs for the institution providing the medical care.

A patient-reported outcome (PRO) is the information provided by a patient or caregiver about what the person is able to do and how that person feels [99,100]. PROs provide what cannot be measured in any way other than from the patient, the family, or the caregiver [101]. 

Patient-reported information about fatigue, anxiety, depression, and sleep disturbance, not only helps the clinician identify side effects of medications [102], but can also help the clinician understand why an increase in chronic recurrent symptoms or disease exacerbation had occurred recently [103]. PROs have the potential “to improve quality of care and disease outcomes, provide patient-centered assessment for comparative effectiveness research, and enable a common metric for tracking outcomes across providers and medical systems” [104]. This will be possible when PROs are standardized [105,106]. Unfortunately, we are not there yet. The implementation of PROs into clinical care has been facilitated by their documented validity, the rapid progression of technology, and greater demand for measurement and monitoring of PROs by regulators (especially the FDA), payers, accreditors, and professional organizations [107]. 

e-PRO is a term used by some to indicate electronic-entered PROs, which provide information comparable to that provided by questionnaires completed on paper [108,109,110], and have the added benefit of not requiring separate data entry. Sometimes the entry is at the time of the visit with the clinician, either via a kiosk [111] or a tablet [112]. When this is completed before seeing the clinician, the information can be available just before or during the visit. 

We recommend entry of e-PROs in real-time and remotely. These e-PROs can then be used to screen for deterioration of adverse events [105,113], and can identify clinically-significant events or changes [114]. For example, patients or their caregivers can provide asthmatic attack information in real-time that can readily be entered into the EHR [115]. This information can prompt intervention when indicated (v.i., alerts) [116]. Conversely, when all is well, information about the absence of changes and deterioration can justify delaying the next visit, thereby helping to reduce the frequency of (unnecessary) outpatient visits [64,117]. 

With physicians increasingly accepting the value of ePROs [118,119,120,121], the data collected routinely can include measures of satisfaction with care, and drug adherence [122]. PROs also appear to contribute to improved communication and diagnosis/treatment, and to a lesser extent, more favorable outcomes [19,105,123]. Some PROs are particularly effective at identifying mental health conditions [124,125].

The systematic monitoring of the health-related quality of life (HRQoL) is being increasingly encouraged [117,126,127,128,129,130]. HRQoL tools have the potential to improve patient–physician communication, “strengthen information exchange” and improve care [131]. In one study of routine cancer care, patients randomized to provide ePROs had a higher HRQoL than the comparison group, and also had a modestly (6%) higher 12-month overall survival, and a modestly lower (7%) rate of admission to the emergency room [132]. In another study of adult cancer patients who completed PROs, those who had the opportunity to discuss the PRO contents with the clinician, subsequently had higher HRQoLs than those deprived of this opportunity [133]. However, not all HRQoL instruments are equivalent, prompting a call for “greater consensus of content across different HRQoL instruments” [134]. This is not easily achieved in light of the propensity to use “disease-specific” quality of life measures [135]. Diaries tracking the occurrence of medical events and treatments are also a form of patient-reported outcome [136]. 

The National Heart, Lung, and Blood Institute’s Expert Panel Guidelines for the Diagnosis and Management of Asthma recommends that clinicians should encourage patients who have asthma (and their families) to use self-assessment tools of asthma control, such as a daily diary [137]. Electronic diaries for entry of symptoms and pulmonary functions, are not only acceptable, but are also helpful in improving control [138,139,140,141,142]. 

Diaries, especially electronic diaries, can be added to the her or even entered directly. They exemplify our goal of having the patient (or caregiver) enrich the EMR, without an intermediary filtering out and perhaps inappropriately changing what was provided as truth.

Patient-generated health data (PGHD) are defined as “health-related data including health history, symptoms, biometric data, treatment history, lifestyle choices, and other information-created, recorded, gathered, or inferred by or from patients or their designees (i.e., care partners or those who assist them) to help address a health concern” [143]. PGHD differ from e-PROs, in including sensor- or device-generated data [144,145], and not requiring that the data be entered directly into the EMR. On the other hand, having the patient enter his/her own data into the EMR, offers the promise of improved collaboration between the patient and the provider [146], and improved identification of adverse drug reactions [147].

### 3.3. Borrowing from a Slightly Older Model: The Medical Home

The term “medical home” was first used to describe the coordination of, and communication among, multiple sources of medical care for children with complex medical needs [148,149]. Although the terms “medical home” and “patient-centered medical home” (PCMH) continue to be used, “care coordination” [15,150,151,152,153], “integrated care” [154,155,156,157], and “medical neighborhood” [158,159,160,161] also appear to be used frequently. Both pediatricians [150,162] and internists [163] embrace the patient-centered medical home for patients with complex needs. Nevertheless, adoption rates are less than desired [154,157,164,165].

The shared characteristics of these concepts include coordination of care among professionals, facilities, and support systems; continuity of care (i.e., between office/clinic visits); modification of care to meet each patient’s (and family’s) needs and preferences; and sharing of responsibility between patient and caregivers to achieve the common goals of coordinated care [166]. Each of these components is a part of the system of medical care we expect will apply to the delivery of care in accountable care organizations.

Low provider reimbursement has been offered as one reason for the low adoption of coordinated care management practices [153,159,167]. Provider reimbursement is also likely to influence how well the “new patient care model” is adopted and implemented. Also likely to influence the “new patient care model”, is the evidence that information technologies might be able to enhance efforts to achieve care coordination [151,152].

### 3.4. Acceptance of New Technologies

The acceptance of information technology can be viewed in light of four constructs, performance expectancy, effort expectancy, facilitating conditions, and social influences [168]. Among the facilitating conditions are, improved electronic systems, flexibility of collection location, and integration with patient health care data elements [107,169]. Older children, adolescents, caregivers, and clinicians all viewed PROs as having the potential to alter the scope of clinical discussions favorably, although not always for the same reasons [170,171,172].

Physicians have shown some hesitance about implementing e-health, because of potential overload due to the additional work of reading emails and reviewing other electronic data, and the need to detail the EMR during the patient’s visit might degrade the patient–physician relationship [173]. Then again, how will reimbursement deal with the increased workload? How will data safety and patient privacy be preserved?

### 3.5. Remote Surveillance 

We use the terms ‘remote surveillance’ and ‘remote monitoring’ for the collection of information from the patient when she/he is not in the office/clinic or in the hospital. Others use such terms as ‘remote patient monitoring’ [174], ‘remote monitoring of patient self-care’ [175], and ‘lifestyle monitoring’ [176].

Monitoring can be continuous and non-obtrusive, as exemplified by the monitoring of patients who have an implanted ventricular pacemaker. Monitoring can also be intermittent and very intense (daily telephone report of weight and symptoms, supplemented by reports from home-care services and individualized home visits by a clinician) [177], or it can be intermittent and less intense when the patient’s complete their own diaries and other ePROs. Remote surveillance has helped reduce mortality, visits to the emergency department, and hospital admissions [65,177,178,179,180,181,182].

The new goal is reducing the need for visits to the office/clinic [183]. With sufficiently-frequent patient- or caregiver-provided reports, the clinician can decide whether a patient needs an outpatient visit, medication adjustment, or watchful waiting, thereby, allowing more flexible, targeted, and cost-saving chronic-disease care [7,184,185,186]. Indeed, clinicians seem to value timely electronic patient-reports of symptoms and other characteristics/events [187,188,189,190]. Patients also appreciate the telemonitoring because it provides reassurance of continuous practitioner surveillance [191].

### 3.6. Alerts 

The natural history of every disease can be viewed as comprised of a continuum of multiple steps, along a trajectory of progression. At each stage, the probability of moving to the next stage can vary at any time [192]. Machine-learning architecture can systematize the procedures needed to assign patients to the most appropriate level of surveillance [58,193]. 

Electronic alerts are intended to call attention to the patient’s increasing probability of moving to the next step along the path, to more severe symptoms. The presumed value of such alerts is that if the clinician intervenes, she/he might be able to limit progression to the next stage. This might hold for episodic disorders, such as asthma, diabetes, and epilepsy [194], just as it might hold for acute kidney dysfunction [195,196,197], and cancer [8,198,199,200,201]. 

## 4. The Future

We suggest that the four-components of the new health-care systems we envisioned, (i.e., increased self-management, reliance on electronic patient-reported outcomes, remote surveillance of patient-provided communications, and systems that alert medical care providers when intervention might be appropriate) have a bright future. Yes, obstacles will occur along the way, with more road-blocks than simple bumps. On the other hand, the economic [202,203] and social pressures (e.g., increasing use of electronic technology) [204,205], to increase patient empowerment [206,207,208]) are likely to be insurmountable.

We also envision the sharing of ePROs (and other relevant data contained within a medical-care organization) with an “information commons” consisting of networks that are prepared to share such information [209,210,211,212]. This will allow the clinician to query if patients similar to theirs also have insomnia since they have started a new medication.

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
