# Peer review of "Characteristics of Future Models of Integrated Outpatient Care"

_healthcare, 2019, doi:10.3390/healthcare7020065_

Round 1

Reviewer 1 Report

The authors present an important review-type study on a relevant and current topic.

Even so, the authors should consider some considerations below:

1. Consider that the authors argue that in their study they describe how patients, their families (and caregivers) can work with members of the healthcare team to achieve these two goals: maintain (and perhaps improve) high-quality care and minimize costs.

In the manuscript, the authors do not address in depth the objectives set: parameters for measuring the quality of care and cost evaluation.

2. Although the authors argue that current technological capabilities allow greater reliance on the use of mobile technologies. This argument is not discussed in depth.

3. Some references do not retain the format of the journal [15, 35]. And others are too old for this review [11, 12]

4. In the Introduction section

Paragraph 3 (line 40 to 47) without references.

5. Authors should consider writing some conclusions of the study-type review.

Author Response

Below, I identify each recommendation/suggestion in blue italics, followed by my response in black typeface.

Reviewer # 1

1. The authors do not address in depth the objectives set: parameters for measuring the quality of care and cost evaluation

Our very second paragraph has these two sentences:

The multiple initiatives to encourage innovations in medical-care delivery that both enhance quality of care and reduce costs have allowed considerable flexibility [1-5].  Here, we offer one ground-level perspective of clinicians and other care providers who are preparing for the soon-to-be reality of outpatient care of children and adults who have chronic diseases.

Thus, we make clear at the very beginning of the manuscript that “we offer one ground-level perspective.” We did not set any objective to measure “the quality of care and cost evaluation.” We did not intend to “address in depth” the objectives that the reviewer set. We intended to describe one innovation “in medical-care delivery that both enhance(s) quality of care and reduce(s) costs.

We can readily understand the reviewer’s disappointment. My colleagues and I have reviewed manuscripts for journal editors and were disappointed that the authors did not write the manuscript we had hoped to read. In these situations, we do not encourage the authors to rewrite their manuscripts. Rather, we encourage them to write such a new manuscript in the near future.

Perhaps some of what the reviewer seeks can be found in:

quality of care (as measured by the patient)

Male L, Noble A, Atkinson J, Marson T. Measuring patient experience: a systematic review to evaluate psychometric properties of patient reported experience measures (PREMs) for emergency care service provision. Int J Qual Health Care. 2017 Jun 1;29(3):314-326.

cost-related issues

Cookson R, Mirelman AJ, Griffin S, Asaria M, Dawkins B, Norheim OF, Verguet S, J Culyer A. Using Cost-Effectiveness Analysis to Address Health Equity Concerns. Value Health. 2017 Feb;20(2):206-212.

2. The authors do not discuss “how current technological capabilities allow greater reliance on the use of mobile technologies.”

True. On the other hand, we did not mention anywhere, “reliance on the use of mobile technologies.” Here, too, it appears that the reviewer wished that we had done so.

Perhaps some of what the reviewer seeks can be found in:

General

Bhavnani SP, Narula J, Sengupta PP. Mobile technology and the digitization of healthcare. Eur Heart J. 2016 May 7;37(18):1428-38

Sharp M, O'Sullivan D. Mobile Medical Apps and mHealth Devices: A Framework to Build Medical Apps and mHealth Devices in an Ethical Manner to Promote Safer Use - A Literature Review. Stud Health Technol Inform. 2017;235:363-367.

By specialties

Maternal and child health

Tamrat T, Kachnowski S. Special delivery: an analysis of mHealth in maternal and newborn health programs and their outcomes around the world. Matern Child Health J. 2012 Jul;16(5):1092-101.

Chronic obstructive lung disease

McCabe C, McCann M, Brady AM. Computer and mobile technology interventions for self-management in chronic obstructive pulmonary disease. Cochrane Database Syst Rev. 2017 May 23;5:CD011425.

Diabetes mellitus

Shan R, Sarkar S, Martin SS. Digital health technology and mobile devices for the management of diabetes mellitus: state of the art. Diabetologia. 2019 Apr 8. [Epub ahead of print]

Gastroenterology

Spiegel B. 2015 American Journal of Gastroenterology Lecture: How Digital Health Will Transform Gastroenterology. Am J Gastroenterol. 2016 May;111(5):624-30.

Behavior change

Short CE, DeSmet A, Woods C, Williams SL, Maher C, Middelweerd A, Müller AM, Wark PA, Vandelanotte C, Poppe L, Hingle MD, Crutzen R. Measuring Engagement in eHealth and mHealth Behavior Change Interventions: Viewpoint of Methods. J Med Internet Res. 2018 Nov 16;20(11):e292

3. Some references do not retain the format of the journal [15, 35]. And others are too old for this review [11, 12]

We have made the corrections to references 15 and 35, and were unaware that references become too old. We would like to retain these old references, in large part because of their age.

4. In the Introduction section, paragraph 3 (line 40 to 47) without references

The last line of paragraph 3 is “In this overview, we discuss each of these elements, first individually, and then about systems that integrate them.” Each of the items listed in this paragraph has multiple references later in this paper. Thus, we did not consider references needed here.

We were not aware that every paragraph was supposed to have at least one citation. This paragraph, however, does have a reference (#6) to a paper about the broad topic of “the future of medicine.’

5. Authors should consider writing some conclusions of the study-type review.

We see the last 2 paragraphs that constitute what we head as “THE FUTURE” as providing the conclusions we consider most appropriate. We could label this section CONCLUSIONS, but here, too, we would much prefer to leave the heading as it is.

Reviewer 2 Report

More guidelines description than a scientific work.

Minor style typos present in the paper (i.e. in raw 74 some words bigger than others).

Paper long 17 pages. Only 7 for description, other for reference. But It is a Review type of publication.

Work suitable for publication in Healthcare for MDPI. 

Author Response

Below, I identify each recommendation/suggestion in blue italics, followed by my response in black typeface.

1. More guidelines description than a scientific work.

We appreciate that the reviewer recognizes what we wanted to achieve

2. Minor style typos present in the paper (i.e. in raw 74 some words bigger than others).

We tried again to find discrepancies in typeface and the only consistent typeface discrepancies are a result of the EndNote program for Healthcare that enters citation numbers in 10 point while allowing the remainder of the manuscript to be 12 point. 

3. Paper long 17 pages. Only 7 for description, other for reference. But It is a Review type of publication.

We are glad the reviewer recognized this intentional discrepancy. We wanted to have a tightly-written, easy-to-read, not-too-long manuscript that provided abundant support for each of the important points we wanted to make.

4. Work suitable for publication in Healthcare for MDPI.

Delighted the reviewer agrees.